# Management of Obesity-Related Inflammatory and Cardiovascular Diseases by Medicinal Plants: From Traditional Uses to Therapeutic Targets

**DOI:** 10.3390/biomedicines11082204

**Published:** 2023-08-05

**Authors:** Bashar Saad

**Affiliations:** 1Al-Qasemi Academic College, Baqa Algharbiya 30100, Israel; bashar@qsm.ac.il or bashar.saad@aaup.edu; 2Department of Biochemistry, Faculty of Medicine, The Arab American University, Jenin P203, Palestine

**Keywords:** inflammation, obesity, diabetes, CVDs

## Abstract

Inflammation is a crucial factor in the development and progression of cardiovascular diseases (CVD). Cardiac remodeling in the presence of persistent inflammation leads to myocardial fibrosis and extracellular matrix changes, which reduce cardiac function, induce arrhythmias, and finally, cause heart failure. The majority of current CVD treatment plans concentrate on reducing risk factors such as hyperlipidemia, type 2 diabetes, and hypertension. One such strategy could be inflammation reduction. Numerous in vitro, animal, and clinical studies indicate that obesity is associated with low-grade inflammation. Recent studies have demonstrated the potential of medicinal plants and phytochemicals to cure and prevent obesity and inflammation. In comparison to conventional therapies, the synergistic effects of several phytochemicals boost their bioavailability and impact numerous cellular and molecular targets. Focusing on appetite, pancreatic lipase activity, thermogenesis, lipid metabolism, lipolysis and adipogenesis, apoptosis in adipocytes, and adipocyte life cycle by medicinal plants and phytochemicals represent an important goal in the development of new anti-obesity drugs. We conducted an extensive review of the literature and electronic databases, including Google Scholar, PubMed, Science Direct, and MedlinePlus, for collecting data on the therapeutic effects of medicinal plants/phytochemicals in curing obesity and its related inflammation and CVD diseases, including cellular and molecular mechanisms, cytokines, signal transduction cascades, and clinical trials.

## 1. Introduction

Inflammation is a physiological response of the body to injury or infection, representing a universal reaction to various local tissue changes [1,2]. It is characterized by a set of standard vascular changes that result in swelling, followed by the movement of white blood cells to the affected area, forming an inflammatory focus [3]. The primary goal of inflammation is to identify the damaging element, remove it, and then regenerate or repair the affected tissue [4].

Cytokines are important signaling proteins involved in the inflammatory host response to inflammatory stimuli. They are classified as either pro-inflammatory (e.g., tumor necrosis factor-alpha (TNF-α), interleukine (IL)-1 (IL-1), IL-6, IL-15, IL-17, and IL-23), or anti-inflammatory (e.g., transforming growth factor beta(TGFβ), and interferon (IFN), IL-4, IL-10, and IL-13) [5]. Chronic and persistent inflammation, as well as auto-immune response, have been linked to atherosclerosis, myocardial infarction, asthma, diabetes, psoriasis, osteoporosis, angiotensin II-derived hypertension, tumor progression, and cardiovascular disease (CVD) [1,2,6,7]. The latter, the world’s largest cause of morbidity and mortality, includes hypertension, heart diseases, peripheral vascular disease, heart failure, and cardiomyopathies [6,7].

Inflammation has been identified as the primary initiator of CVD etiology [8]. The immune response and tissue repair are coordinated by the interactions between inflammatory cytokines. The length of the inflammatory state and the continuous activation of pro-inflammatory pathways may be associated with a slower healing process [9,10]. CVD increases the levels of several mediators, including C-reactive protein (CRP), pentraxin-related protein (PTX3), and TNF-α [11,12,13]. Atherosclerosis, produced by endothelial dysfunction and increased permeability of low-density lipoprotein (LDL) and immune cells into the intima [14], is widely recognized as a primary contributor to a wide range of CVDs [15]. Surprisingly, the link between inflammation and atherosclerosis is important in clinical cardiovascular risk, generating interest in anti-inflammatory (Figure 1). 

Because the number and kinds of secondary metabolites in a single herb vary [16], the pharmacological effects of herbs are specific to certain plant species or groups of plants. Secondary metabolites do not play a part in normal cell/tissue growth, development, and reproduction; rather, they serve as a defensive factor to protect a plant from potential environmental and interspecies injury. As a result, secondary metabolites are often produced by plants to regulate their metabolism in response to the local presence of herbivores, pollinators, and microorganisms. These metabolites are often derived from primary metabolites or share substrates with them and are produced to meet specific needs. Thousands of these secondary metabolites have been found, and the list is growing. Secondary metabolites are broadly classed as alkaloids, terpenoids, and phenolics [16,17,18] (Figure 2).

## 2. Medicinal Plants and Their Anti-Inflammatory Mechanisms 

Low-grade chronic inflammation in adipose tissue and throughout the body is an important factor in the development and progression of CVD and other obesity-related chronic illnesses. Pre-adipocytes, mature adipocytes, fibroblasts, and infiltrating macrophages are found in both brown and white adipose tissue [19]. Adipocytes produce leptin, adiponectin, and chemokines that can affect inflammation and promote tumor growth. While adipose tissue typically secretes anti-inflammatory mediators such as adiponectin, an increase in fat mass can cause the secretion of pro-inflammatory cytokines like TNF-α, IL-1, IL-6, and MCP-1, as well as leptin [19,20] (Figure 3).

Augmented fat metabolism raises free fatty acid levels, which trigger inflammatory signaling cascades in infiltrating macrophages, dendritic cells, and T lymphocytes. Pro-inflammatory cytokines reinforce via paracrine and autocrine mechanisms the inflammatory state and drive macrophage infiltration and activation [19,20,21]. The potential therapeutic benefits of medicinal plants/phytochemicals derive from the synergistic effects of numerous phytochemicals, which boost bioavailability and activity on many cellular and molecular targets. Many plants and their active compounds have been demonstrated to reduce inflammation in fat tissue [22,23,24,25,26,27,28]. As a result, there is a lot of interest in finding plant-derived compounds for developing effective and safe anti-inflammatory medicines [29,30]. Several clinical studies have demonstrated that curcumin can reduce inflammation in obese individuals by restoring the balance between pro-inflammatory and anti-inflammatory mediators. This is achieved through its interaction with various cellular components, including transcription factors, receptors, growth factors, enzymes, cytokines, and chemokines [29,30]. A meta-analysis of 16 randomized controlled trials RCTs including 1010 people found that ginger had a significant effect on lowering circulating C-reactive protein (CRP), high sensitivity C-reactive protein (hs-CRP), and TNF-α levels. Large-scale randomized controlled trials (RCTs) are still needed to draw strong findings about the impact. Studies have shown that ginger can reduce levels of circulating C-reactive protein (CRP), high sensitivity C-reactive protein (hs-CRP), and TNF-α. However, large-scale randomized controlled trials (RCTs) are still needed to confirm its effects on other inflammatory mediators [31]. According to a meta-analysis of 20 clinical trials, pomegranate and its active compounds reduce BMI, hypertension, blood glucose levels, triglycerides, total cholesterol, and LDL. It may help reduce insulin resistance and raise HDL levels. Despite the fact that meaningful effects have been seen, more well-designed clinical trials are required to discover the optimum formulations and doses that may be utilized to prevent or treat MetS components [32].

The biological activities of anti-obesity and anti-inflammatory medicinal plants/phytochemicals, as well as their impact on CVDs, are reviewed here.

TNF-α is a significant inflammatory mediator among cytokines, with many effects on inflammatory cells, endothelial cells, and fibroblasts [33]. Multiple inflammatory stimuli activate macrophages and T cells, which then release TNF-α, which acts through a positive autocrine mechanism, resulting in increased TNF-α secretion as well as other cytokines such as IL-8 [33]. Abnormal TNF-α secretion, on the other hand, mediates a variety of diseases such as chronic inflammatory diseases, autoimmune diseases, and other chronic diseases through the secretion of additional pro-inflammatory factors [34], which can cause direct DNA damage and play an apoptotic or anti-apoptotic role depending on the signaling promoters [33,34].

TNF-α operates through two receptors: TNFRI, which is found in the majority of the body’s cells, and TNFRII, which is found largely in hematopoietic cells. TNFRI activation induces the gene expression of COX-2, IL-1, IL-6, and matrix metalloproteinase (MMP), chemokines, and adhesion molecules via signal transduction cascades (e.g., FAS-associated signal transduction through death domain, mitogen-activated kinase (MAPK), Jun kinase (JNK)/activation protein-1 (AP-1), receptor-interacting protein (RIP) 3, and IKK-NF-B). Caspase3/caspase8 activation can cause apoptosis, and receptor-interacting protein (RIP) 3 activation can cause necrosis [35]. TNFRII activation generates a variety of inflammatory mediators and growth factors via the transcription factors AP-1 or NF-B. This cascade activates apoptosis-inducing negative regulators such as Bcl-2 (B-cell lymphoma 2) and superoxide dismutase. Kumar and his colleagues [36] have discovered that TNFR1 is involved in the activation-induced death of pathological CD4+ T lymphocytes in ischemic heart failure. They observed that CD4+ T cells become pathological during heart failure and that the expression of TNF-α and TNFR1 increases in these cells. The role of the TNF-α/TNFR1 axis in T-cell activation and proliferation is not yet known. When TNFR1 is neutralized during T-cell activation or when HF-activated CD4+ T cells are transferred without TNFR1, their survival and proliferation signals are enhanced. Interestingly, TNFR1 neutralization does not affect the expression of CD69 or the pathological activity of these cells. These findings suggest that TNFR1 plays a key role in suppressing survival and proliferation signals in CD4+ T cells during heart failure without changing their pathological activity.

Interleukins (IL) are a key class of cytokines that play a crucial role in immune response. One of these, IL-1, is involved in the transformation of phagocytes during inflammation or cancer. It helps produce reactive oxygen (ROs) and nitrogen species (Nos), as well as inflammatory molecules such as chemokines, integrins, and MMPs [37]. IL-6 is another key molecule in acute inflammation, and its uncontrolled production leads to a variety of inflammatory diseases [38]. IL-6 is primarily produced by monocytes, macrophages, and T cells. Its production and release are regulated by several transcription factors, including NF-κB and AP-1 [38]. When IL-6 binds to its receptor (IL-6R), it causes two gp130 chains to come together and activates the Janus kinases (JAKs) attached to them. The JAKs then add phosphate groups to the gp130, which attracts and activates several other molecules, including the transcription factors STAT3 and STAT1, as well as PI3K, Ras-MAPK, and SHP2. [39,40]. 

Many plant extracts were found to reduce the production levels of IL-6 and TNF-α using cultured human cell lines. *Hypericum triquetrifolium*, for example, attenuated the mRNA and protein of TNF-α and IL-6 in lipopolyssacharide-stimulated human THP-1 cells and human peripheral blood mononuclear cells (PBMNCs). *Hypericum triquetrifolium* significantly reduced the expression and secretion of both cytokines. These findings indicate that *Hypericum triquetrifolium* -extract probably exerts anti-inflammatory effects at both protein and gene expression levels of the anti-inflammatory as well of the pro-inflammatory cytokines in PBMNCs [41,42]. Similar results were reported with extracts from *Peganum harmala* [43], *Salvia officinalis, Alchemilla vulgaris*, *Rosmarinus officinalis, Eriobotrya japonica, Olea europaea*, [44,45], *Arum palaestinum, Ocimum basilicum, Trigonella foenum-graecum* [46], *Ficus sycomorus* [47], *Inula helenium, Saponaria officinalis* [48], Cucumin, *Camellia sinensis, and Zingiber officinale* [49,50]. 

Transcription factors control the gene activity of several pro-inflammatory mediators. NF-B, STAT1/STAT3, AP-1, Nrf2, and HIFs are all involved. Both NF-κB and JNK) were discovered to be crucial in the activation of pro-inflammatory genes downstream of TLRs in adipose tissue. Hence, targeting the TLR4/NF-κB or TLR4/JNK will inhibit inflammatory responses in adipose tissue and associated insulin resistance in obese patients [51,52,53,54,55,56,57]. NF-κB, the most focal transcription factor, comprises two subunits known as NF-κB/Rel complex and is found as an inactive complex form [51]. When the body is exposed to pro-inflammatory cytokines (like IL-1 and TNF-α), viruses, or lipopolysaccharides, it activates NF-κB. This happens through the breakdown and phosphorylation of IkBα, which allows NF-κB to move to the nucleus and bind to the promoter region of genes that produce pro-inflammatory mediators like COX-2, iNOS, and cytokines [52]. NF-κB plays an active role in chronic inflammatory conditions such as Crohn’s disease, inflammatory bowel disease (IBD), and inflammatory lung and kidney diseases. [53,54,56,57]. Many plant extracts were found to inhibit NF-κB signaling. For example, *Zingiber officinale, Calea urticifolia, Solanum lycopersicum, Angelica sinensis, and Nigella sativa inhibit* the inflammatory process by inhibiting the NF-κB pathway [58,59]. *Nigella sativa* extracts were found to significantly increase the production of NO, COX-2, TNF-α, and IL-6 by activating the JNK, ERK1/2, P38, and NF-κB pathways [58,59]. 

The anti-inflammatory effects of Curcumin, *Forsythiae fructus, Camellia sinensis, and Tripterygium wilfordii* is mediated, at least partially, by affecting the JAK-STAT signaling pathway [49,50]. Curcumin inhibits the JAK-STAT cascade and blocks the synthesis of IL-12, thus relieving inflammation [29,30]. In addition, it inhibits the activation of STAT1 and STAT3 and depresses pro-oncogenic inflammatory pathways, including nuclear factor-κB (NF-κB) and interleukin-6 (IL-6), and JAK pathways [38].

Reactive nitrogen species (RNS) include nitrogen dioxide, peroxynitrite, and other forms. The activation of nitric oxide synthases (NOSs) results in the production of nitric oxide (NO), which serves as a cellular or intracellular signaling molecule. NO can create or change intracellular signals at physiological quantities, which can have an impact on the activity of immune cells, tumor cells, and resident cells in many tissues and organs [60]. However, when an infection occurs, its high levels might lead to the destruction of the target tissue. Endothelial NOS (eNOS), neuronal NOS (nNOS), and inducible NOS (iNOS) are the three distinct isoforms of NOS that have been identified. The latter kind is an inducible enzyme that is substantially produced by inflammatory stimuli in some cells, including macrophages, as opposed to the first two, which are constitutively expressed in the body. A considerable increase in the quantity of NO generated by iNOS contributes to the inflammatory process and works synergistically with other inflammatory mediators. Inhibiting iNOS activity or decreasing iNOS expression may help to lessen the inflammatory response. As a result, it is worthwhile to assess the inhibitory effects of plant extracts on NO generation [61]. In one of our earlier investigations, we looked at the effects of *Hypericum triquetrifolium* extracts on NOS activity [42]. Results obtained in this study show for the first time that *Hypericum triquetrifolium* down-regulates the production of NO by inhibiting the transcription of the iNOS gene. Similar findings were made using therapeutic plant-derived substances (for example, flavonoids). Flavonoids such as apigenin, luteolin, and quercetin were reported to suppress NO generation in LPS/cytokine-treated macrophages or macrophage-like cell lines. Mechanism investigations, however, revealed that flavonoids did not significantly suppress iNOS. They were shown to down-regulate iNOS induction, resulting in decreased NO production [62]. Moutan Cortex, a traditional medicine used to remove heat from the blood, promote blood circulation, and relieve blood stasis, has been found to have anti-inflammatory effects by suppressing the phosphorylation of I-kBa and activation of NF-kB [63]. Furthermore, many plant extracts were found to reduce the production levels of NO using cultured human cell lines. For example, in previous studies, we found that *Rosmarinus Officinalis, Eriobotrya Japonica, Olea europaea* [44], *Ficus sycomorus* [47], *Arum palaestinum, Ocimum basilicum, Trigonella foenum-graecum* [46], *Inula helenium, Saponaria officinalis* [48], and *Peganum harmala* [43]. 

IL-10 is a versatile anti-inflammatory and immunosuppressive cytokine produced by various immune cells, including dendritic cells, macrophages, mast cells, natural killer cells, eosinophils, neutrophils, B cells, cytotoxic T cells, and several types of T helper cells. Its production is regulated by both direct and indirect stimulation of the stress axis. Dysregulation of the stress axis by IL-10 can lead to numerous inflammatory diseases such as neuropathic pain, Parkinson’s disease, Alzheimer’s disease, osteoarthritis, rheumatoid arthritis, psoriasis, systemic lupus erythematosus, type 1 diabetes, inflammatory bowel disease, and allergies Natural anti-inflammatory compound curcumin has the ability to increase IL-10 synthesis and expression while also enhancing its effects on a range of tissues. In vitro and in preclinical models, curcumin is able to influence the pathophysiology of diseases such as pain and neurodegeneration [64].

In one of our previous studies, we evaluated the effects of extracts from *Hypericum triquetrifolium* on NOS [42]. PBMNCs treated with *Hypericum triquetrifolium* extract had significantly higher levels of IL-10 protein and mRNA expression. PBMNCs were treated with LPS, which resulted in the release of a modest amount of IL-10. The extract of *Hypericum triquetrifolium* elevated IL-10 twofold. Other studies have convincingly demonstrated that IL-10 inhibits the release of pro-inflammatory cytokines [64,65,66], suggesting that *Hypericum triquetrifolium*-extract-mediated inhibition of the LPS-induced secretion and mRNA expression of IL-6 and TNF-α may pass through the induction of IL-10 production. Several inflammatory disorders have both a very low blood level of IL-10 and a high blood level of TNF-α. Furthermore, injections of the recombinant version of IL-10 reduced TNF-α levels in the blood, which has been shown to be advantageous in such conditions. [66,67,68]. The ability of *Hypericum triquetrifolium*-extract to modulate both pro-inflammatory and anti-inflammatory cytokines in LPS-activated PBMNCs adds to the case that it could be used as an alternative or supplement in the treatment and/or prevention of inflammatory diseases. The discovery that varied extract concentrations impact the production of pro-inflammatory and anti-inflammatory cytokines suggests that more research is needed.

## 3. Anti-Obesity Medicinal Plants and the Molecular Mechanisms Underlying Their Activities

Currently, medicinal plants, unprocessed extracts, and phytochemicals that aim to regulate weight are becoming more widespread. Significant decrease in body weight was seen by *Nigella sativa (black seeds)*, *Curcuma longa* (curcumin), *Cinnamomum verum* (cinnamon), *Panax ginseng* (ginseng), *Foeniculum vulgare, Phaseolus vulgaris, Camellia sinensis* (green tea), *Allium sativum* (garlic), *Crocus sativus*, and virgin olive oil (Figure 4). These plants have been used in various traditional medical systems for a long time, and their anti-obesity and antidiabetic effects have been confirmed in numerous in vitro animal and clinical testing [28,69,70,71,72]. 

There has been a lot of progress in understanding how phytochemicals can reduce weight, according to a number of lines of evidence. One of the main categories of plant secondary metabolites is polyphenols, which are abundantly present in medicinal plants as well as in fruits, vegetables, grains, and legumes [73]. Their anti-obesity abilities have been repeatedly demonstrated in cell culture, animal, and clinical research. Examples of substances that have been shown to have weight-reducing properties include genistein and diadzein, cyanidin, grape seed proanthocyanidin extract, xanthohumol, apigenin and luteolin, kaempferol, myricetin and quercetin, and (-)-Epigallo-cathechin gallate (EGCG). Likewise, phytosterols, polyunsaturated fatty acids (PUFA), and organosulfur compounds are other bioactive dietary ingredients having anti-obesity characteristics [74,75,76,77,78,79,80,81,82,83,84,85,86,87,88]. The primary goals of medicinal plants and the products generated from them include inhibition of pancreatic lipase activity, suppression of hunger, stimulation of thermogenesis and lipid metabolism, prevention of lipolysis, and promotion of adipogenesis (Figure 5). A better treatment method for treating obesity is to target adipogenesis in adipose tissue. With weight reduction, mature adipocytes undergo apoptosis and/or lipolysis, which results in a decrease in adipose tissue mass [74,75] (Figure 5).

### 3.1. Major Basic Anti-Obesity Mechanisms

Targeting appetite: Some anti-obesity phytochemicals that act through appetite reduction include thymoquinone from *Nigella sativa*, saponins from ginseng, HCA from *Garcinia cambogia*, EGCG from green tea, steroidal glycoside from *Hoodia gordonii* and *Hoodia Pilifera*, lectins from *Phaseolus vulgaris* (common bean), and ephedrine from Ephedra species [71,72,75,89]. Appetite and satiety are regulated by a coordinated action of anorexigenic and orexigenic hormones and neuropeptides produced in the hypothalamus as well as in the digestive tract, liver, and adipose tissue [89]. In the short term, neuronal and hormonal transmission from the gastrointestinal tract regulates hunger. The gastrointestinal tract is the biggest endocrine organ in the body and is thought to play a significant role in appetite control through the release of multiple peptide hormones. As a result, these hormones (e.g., ghrelin, melanin-concentrating hormone receptor) might be used to treat obesity. In addition, the inhibition of fatty acid synthase, an adipocyte enzyme, catalyzes the synthesis of fatty acids and represents a potential therapeutic target for appetite suppression. Indeed, many medicinal herbs and their extracts were reported to exert their weight-reducing effect through the inhibition of fatty acid synthase activity and, thus, reduce appetite [74,89].

Targeting pancreatic lipase: Inhibition of fat digestion and absorption is one of the currently used strategies to reduce energy intake through inhibition of the activity of the pancreatic lipase. It is responsible for the hydrolysis of 50–70% of total dietary fats to glycerol and free fatty acids. The latter are then incorporated into bile acid-phospholipid micelles, absorbed in the small intestine, and finally entered the peripheral circulation as chylomicrons [90]. Pancreatic lipase inhibitory effects of medicinal plants and phytochemicals have been extensively investigated. Hitherto, many herbs, as well as isolated phytochemicals, were found to attenuate pancreatic lipase activity. Some phytochemicals that can help reduce weight by inhibiting pancreatic lipase include saponins and catechins from green tea, polyphenols like mangiferin and catechins and condensed tannins from Salacia reticulate, punicalagin, ellagic acid, and anthocyanins from pomegranate, rosmarinic acid and carnosic acid from rosemary, and proteins and isoflavones from soybean [71,72,90,91].

Stimulators of thermogenesis: Thermogenesis is literally defined as heat production. It takes place in the mitochondria of the brown adipose tissue through uncoupling of the mitochondrial electron transport chain by uncoupling protein 1 (UCP1). The “adaptive thermogenesis” that maintains an ideal environment for weight recovery and is active in both lean and obese persons attempting to sustain reduced body weights is what causes the over 80% weight regain to pre-weight loss following a successful weight loss. The adipocyte-derived hormone leptin has a significant role in mediating this resistance to long-term weight loss. Thermogenesis is influenced by phytochemicals and macromolecules in food as well as dietary carbohydrates. Numerous animal and human studies have demonstrated the ability of several phytochemicals to reduce body weight, including oleuropein, capsaicin, resveratrol, epigallocatechin gallate, gingerol, caffeine, and ephedrine [71,72,75,91,92]. Capsaicin, for example, increases catecholamine release from the adrenal medulla to exert its thermogenic action, whereas EGCG promotes thermogenesis by inhibiting catechol methyl-transferase, an enzyme that destroys norepinephrine. Caffeine mediates thermogenic effects by inhibiting the phosphodiesterase-induced degradation of intracellular cAMP, and it decreases energy intake by reducing food intake [71,72,75,91,92].

Increase satiety: Long-term studies have established the favorable weight-loss effects of dietary fiber [93] in obese people, and they are related to a lower BMI. Many entire plant meals include substantial levels of dietary fiber, including pectin, gum, mucilage, cellulose, hemicellulose, lignin, and soluble fibers. Supplementing regular meals with gel-forming fibers, such as guar gum, increases satiation, most likely owing to delayed stomach emptying. Dietary fibers are normally indigestible by humans; however, they may be fermented by gut microorganisms. Soluble or fermentable fibers and insoluble fibers can both be fermented by gut microorganisms to give bulk. Natural hydrogel-forming fibers such as pectin, gum, and mucilage are examples of soluble fibers, whereas structural fibers such as cellulose and lignin are examples of insoluble fibers. Through the hydrogel effect, which slows the absorption of energy-rich macromolecules, insoluble fibers are known to reduce hunger and, thereby, meal intake [71,92,93]. A high-fiber diet decreases both food intake during meals and food intake at the next meal, according to a number of clinical studies. Foods high in pectin cause the stomach to take longer to empty and make you feel fuller [71,92,93]. Recent studies have found a connection between satiation and changes in the hormones orexigenic and anorexigenic. We believe that systematic analyses of the key gut hormone responses to diverse fiber types and formulations will significantly advance our knowledge of this subject. While leaving the HDL fraction alone, hydrogel-forming fibers are particularly effective at lowering high LDL cholesterol. Additionally, they support those with diabetes or impaired glucose tolerance. These effects are most likely connected to the fiber’s gelling characteristic, which slows the absorption process [74].

### 3.2. Targeting Adipocyte Apoptosis

Antiobesity medicinal plants and phytochemicals can reduce the mass of adipose tissue by apoptosis. At the cellular level, obesity is characterized by an increase in the number or size or both of the adipocytes. The adipocyte life cycle consists of alteration of cell shape and growth arrest, clonal expansion, and a complex sequence of changes in gene expression leading to storage of lipids and finally to apoptosis [71,74,94]. During the terminal phase of adipocyte maturation, mRNA levels for enzymes involved in triacylglycerol metabolism (e.g., glyceraldehyde-3-phosphate dehydrogenase, fatty acid synthase, and glycerol-3-phosphate dehydrogenase) rise significantly. While it was previously thought that the number of fat cells in the body stays the same throughout a person’s life, recent evidence suggests that new fat cells can be created and old ones can be removed through a process called apoptosis [74,94].

Targeting preadipocyte apoptosis: Natural compounds that control the adipocyte life cycle have as their main target preadipocyte maturation [95]. It has been demonstrated that several phytochemicals can increase apoptosis and reduce preadipocyte growth. For instance, it has been discovered that the antioxidant activity of flavonoids is related to the triggering of apoptosis in preadipocytes. It has been demonstrated that a number of flavonoids, including naringenin, rutin, hesperidin, resveratrol, naringin, and genistein, have cytostatic effects on preadipocytes. One of the most current flavonoids, quercetin is present in a variety of fruits and vegetables; the green tea polyphenol EGCG, and capsaicin from paprika have been shown to induce preadipocyte apoptosis. Induction of apoptosis in preadipocytes is intermediated by the activation of caspase-3, Bax, and Bak, followed by fractionalization of PARP and downregulation of Bcl- 2. Likewise, treatment of preadipocytes with phenolic acids similar to o- coumaric acid, m- coumaric acid, and chlorogenic acid redounded in cell cycle arrest in the G1 phase in a time- and cure-dependent manner. CLA has recently been found to induce apoptosis in human preadipocytes. During insulin therapy, EGCG also promotes apoptosis in post-confluent premature preadipocytes, although the molecular pathways involved remain unclear. Induction of apoptosis in post-confluent differentiated cells leads to a decrease in the number of adipocytes, as preadipocytes undergo multiple rounds of replication during the first two days of differentiation [74,95,96,97,98,99,100,101,102,103,104,105,106]. The effect of curcumin on the cell cycle was recently determined. It significantly decreased the viability of preadipocytes at a concentration that caused apoptosis in preadipocytes and induced activation of caspases 8, 9, and 3. A non-cytotoxic dose of curcumin (15 µM) inhibited MCE, attenuated the expression of PPARγ and C/EBPα, prevented differentiation medium-induced β-catenin downregulation, and decreased the lipid accumulation in 3T3-L1 adipocytes. These data show that curcumin can induce preadipocyte apoptosis and attenuate adipocyte differentiation, leading to inhibition of adipogenesis [74,95].

Targeting adipocyte apoptosis: Animal studies have shown that EGCG, genistein, capsaicin, soy isoflavones, and CLA reduce body fat. However, the mechanism by which they exert their apoptosis-inducing effects on adipocytes has only recently been investigated. EGCG-induced apoptosis is thought to be mediated by protein 1, nuclear factor kappa B (NF-kB), p53, and caspase-3 activity. Although the effects of conjugated linoleic acid (CLA) on body fat are not fully understood, the marked increase in TNF-α mRNA observed after treatment of adipocytes with uncoupling protein 2 (UCP2) suggests that CLA-induced suspected to be responsible for apoptosis [74,102,103,104,105,106]. Genistein, ajoene, EGCG, and capsaicin exert apoptotic effects by stimulating the release of reactive oxygen species (ROS), thereby activating AMP-activated protein kinase (AMPK), an important target for anti-obesity therapy. Ajoene also induces apoptosis in leukemia cells through the formation of ROS, and more recently, it was shown that ajoene also induces ROS-mediated apoptosis in adipocytes [74,103,104,105,106,107,108,109,110].

Synergistic apoptosis-inducing effects of phytochemicals: The apoptosis regulator Bcl-2 and its homologs, which control the outer mitochondrial membrane, are members of the Bcl-2 family. They are either anti-apoptotic (Bcl-2, Bcl-xL, Bcl-w, etc.) or pro-apoptotic (particularly Bax, Bak, Bok, etc.). The Bcl-2 family of proteins induces the release of cytochrome c and translocates from the cytosol to the mitochondria, where it displays pro-apoptotic action. In 3T3-L1 adipocytes, CLA promotes apoptosis brought on by ajoene. Individually, neither CLA nor ajoene had an impact on cytochrome c, whereas ajoene enhanced and CLA had no impact on the expression of the Bax protein [72,111,112]. However, the expression of cytochrome c and Bax was enhanced when ajoene and CLA were combined. Similarly to this, vitamin D3 increases genistein’s ability to induce apoptosis and prevent adipogenesis in mature 3T3-L1 preadipocytes. Vitamin D3 alone boosted VDR protein levels in 3T3-L1 adipocytes by only 40%, while genistein alone did not enhance these levels at the dosages studied. In contrast, the combination of genistein and vitamin D3 elevated VDR protein levels by more than 100%. With combination therapy, this impact on VDR was associated with a roughly 200% increase in apoptosis. The results of such synergy point to the possibility of reducing the harmful effects of two or more chemicals while still producing desired effects on adipocytes. Despite the fact that outcomes from in vitro and animal experiments cannot be directly extrapolated to clinical efficacy, such studies indicate that selected phytochemicals, alone or in combination, are widely used to treat obesity through adipocyte apoptosis contribute to our understanding of important molecular and cellular pathways. It may be effective in inhibiting adipogenesis [74,111,112,113].

### 3.3. Targeting of Adipogenesis 

Adipogenesis is reduced by medicinal herbs and phytochemicals such as resveratrol, guggulsterone, CLA, capsaicin, baicalein, EGCG, genistein, esculetin, DHA, berberine, and procyanidins. Capsaicin, genistein, berberine, and EGCG inhibited the synthesis of CCAAT/enhancer-binding protein (C/EBP) and peroxisome proliferator-activated receptor (PPAR). These transcription factors have been implicated in the suppression of preadipocyte proliferation, which is essential for adipocyte differentiation. Resveratrol, for example, has been found to decrease adipogenesis by enhancing the expression of the Sirt1 gene, which activates fat mobilization by suppressing PPARγ. Capsaicin, genistein, and EGCG have been shown to inhibit adipocyte differentiation through activation of AMPK [74,94,114]. At the same time, PUFA has been shown to suppress adipogenesis by lowering the expression of sterol-regulatory element-binding proteins and slowing the late phases of adipocyte differentiation. Many cell lines have recently been proven in vitro to undergo lipogenic differentiation into adipocytes. The 3T3-L1 preadipocyte is a well-studied biological model for understanding the adipogenesis process [115,116,117]. To assess the effect of test plant extracts on 3T3-L1 cell adipogenesis, 3T3-L1 preadipocytes are differentiated into mature adipocytes in the presence of different concentrations of the test plants (0–2 days), and the treatment is maintained for a total of 10 days [41,42,72]. Grape seed, green tea, *Allium sativum, Camellia sinensis, Capsicum annuum, Curcuma longa, Ginkgo biloba, Olea europea, Mentha longifolia, Alchemilla vulgaris, Vitis vinifera*, and *Salvia officinalis* extracts [41,42,72] have been reported to reduce adipogenesis in the 3T3-L1 cell line. Salvia officinalis leaves act through synergistic mechanisms to reduce the weight of overweight and obese patients [70]. Further research is needed to determine if combining these extracts could create a more effective way to suppress the formation of fat cells. Adipogenesis, the process of creating new fat cells, is controlled by adipogenic signaling, which activates transcriptional activators, mainly from the PPAR and C/EBP families, to regulate fat cell differentiation and fat storage within cells [115,116]. During the final stages of preadipocyte development, PPAR and C/EBP coordinate the activation of various adipogenic gene products, including ADRP, aP2, CD36, and perilipin. These gene products work together to produce the adipocyte phenotype [115,116].

Lipolysis: Lipolysis in adipocytes and the release of free fatty acids play a major role in regulating energy homeostasis. Hormone-sensitive lipase is the key enzyme that catalyzes lipolysis in adipocytes. Several medicinal plants and phytochemicals can also affect lipolysis [118,119]. Flavonoids genistein, diadzein, coumestrol, and zearalenone stimulate lipolysis in a dose-dependent manner. Quercetin, luteolin, and fisetin increase lipolysis in rat adipocytes in a dosage and time-dependent manner that was synergistic with epinephrine [120,121]. It has also been observed that these flavonoids inhibit phosphodiesterase. Proanthocyanidins produced from grape seeds promote long-term lipolysis in 3T3-L1 adipocytes through elevating cAMP and protein kinase A. In human adipocytes and 3T3-L1 adipocytes, conjugated linoleic acid was observed to increase basal lipolysis. When docosahexaenoic acid (an omega-3 fatty acid) is introduced to mature adipocytes, it also induces lipolysis [74,122,123,124,125]. 

## 4. Concluding Remarks

A major factor in the onset and development of cardiovascular disease is inflammation. Pro-inflammatory mediators and their signaling cascades, such as NF-B, Hippo, and mTOR, cooperate in the interaction between inflammation and cardiovascular disease to regulate cell fate plasticity and mold different biological activities in cardiomyocytes and immune cells.

One of the main strategies for treating obesity and the low-grade inflammation that it causes is weight loss. This can be accomplished by increasing physical activity, consuming fewer calories, and altering behavior. Nevertheless, these strategies fell short. The hunt for secure and efficient herbal products that promote weight reduction is, therefore, a current trend in the treatment of obesity. Future natural medications may be derived from a number of medicinal plants and their active ingredients since they have demonstrated demonstrable usefulness in the treatment of obesity and associated disorders. It has been demonstrated that herbs containing the active ingredients of green tea, *Phaseolus vulgaris*, *Zingiber officinale*, *Garcinia cambogia, Nigella sativa, Punica granatum*, olive oil, turmeric, and *Caralluma fimbriata*, are effective in treating obesity and related conditions.

The drawbacks and difficulties of using medicinal plants and their extracted plant secondary chemicals must be taken into account. The following are the primary restrictions on utilizing these natural products: (1) The repeatability of results obtained using plant extracts is uneven because the type and quantities of secondary plant metabolites rely greatly on environmental factors. This can be decreased by rigorously implementing the standardization and purification of the extracts. While it is widely known that microbial endotoxin can change the immune response, bacterial contamination has not been sufficiently accounted for in the majority of anti-inflammatory research carried out to gauge the effectiveness of medicinal plants. (2) Phytochemists should focus more on creating novel purification techniques to boost the concentration of phytochemicals and natural products for pharmaceutical applications, as their concentration is insufficient for clinical development and usage. (3) Because phytochemicals may have a favorable toxicity profile due to their low availability, caution should be used while developing new delivery methods. Many natural products, diets, medicinal plants, and their active compounds have made considerable strides in their pharmacological benefits, but they are still a long way from being used in clinical settings. Despite the fact that the majority of the research in this review was conducted in cell culture or animal models, there is significant historical, empirical, and scientific evidence that whole plants, not just isolated constituents, have cardiovascular beneficial effects due to anti-obesity, anti-diabetic, and immunomodulatory activities [126,127]. Although there is a lot of information about the positive effects of medicinal plants in preventing and treating cardiovascular diseases (CVDs), their clinical therapeutic benefits have not been established. As a result, herbal treatments cannot be recommended as an alternative therapy. More well-designed studies and clinical trials with larger sample sizes are needed to investigate the role of medicinal plants and phytochemicals in CVDs. Safety and toxicity should also be addressed in future clinical trials.

## Figures and Tables

**Figure 1 biomedicines-11-02204-f001:**
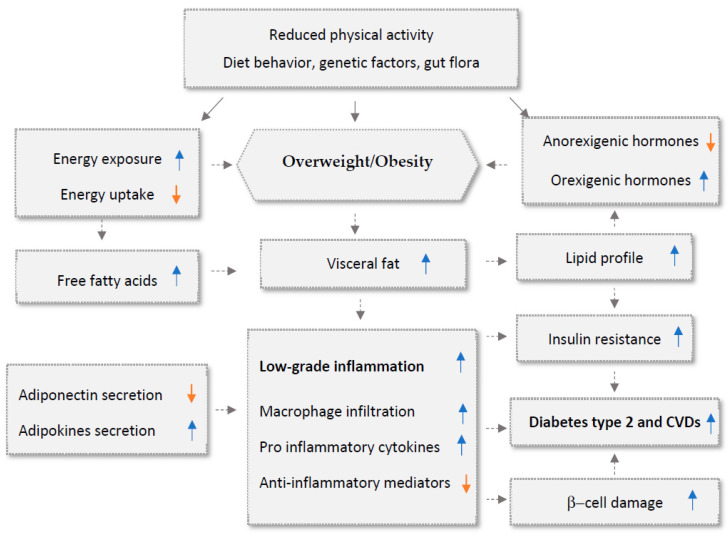
The interconnection between inflammations, type 2 diabetes, and obesity. Up arrows: incresesed effects; down arrows: decresed effects.

**Figure 2 biomedicines-11-02204-f002:**
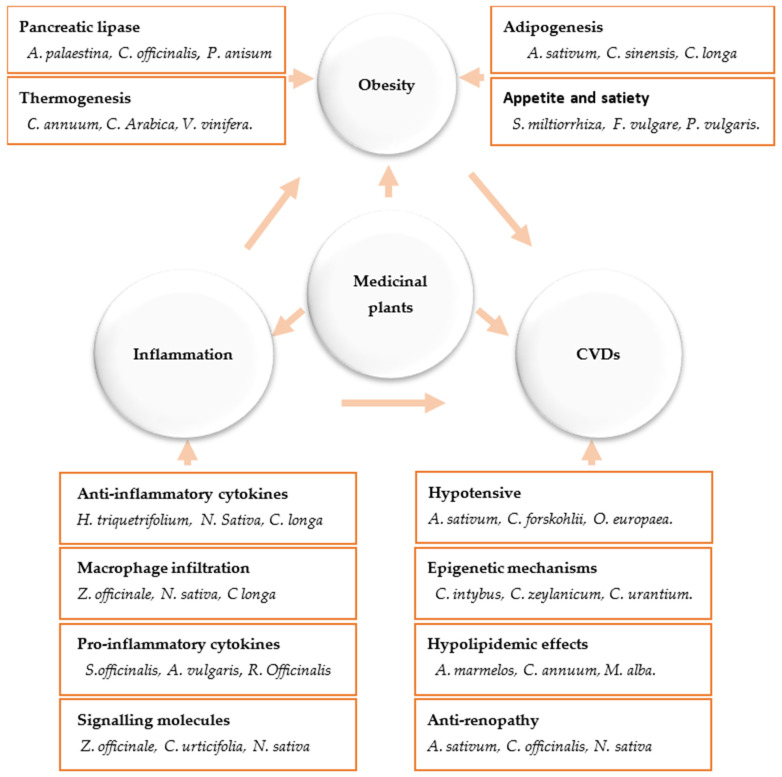
Anti-obesity and anti-inflammatory action mechanisms of medicinal plants and their impact on cardiovascular diseases.

**Figure 3 biomedicines-11-02204-f003:**
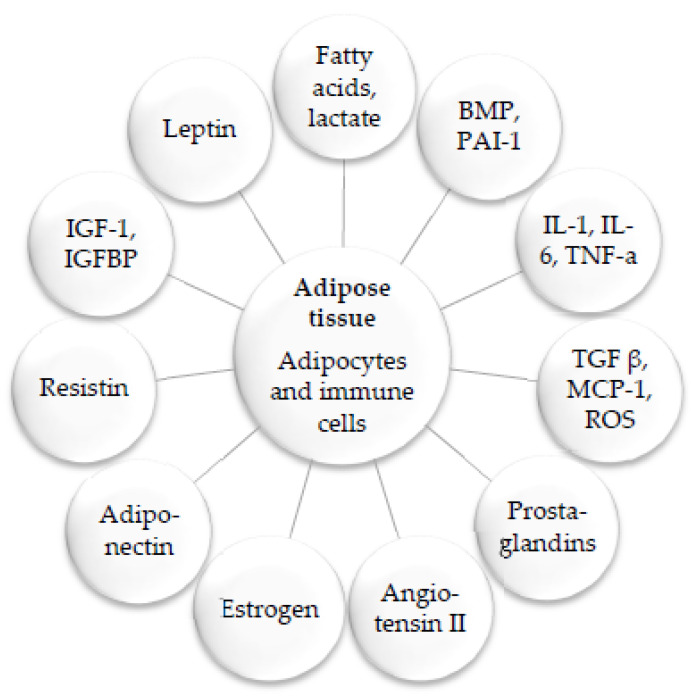
Factors secreted by adipose tissues. Adipose tissues, including brown adipose tissue (BAT) and white adipose tissue (WAT), are composed of mature adipocytes and other smaller cells, such as pre-adipocytes, fibroblasts, and macrophages. These tissues secrete several hormones, including leptin, adiponectin, and resistin. One proposed mechanism for obesity-induced hypertension is the expression of angiotensinogen and other enzymes necessary for its conversion into angiotensinogen II in adipose tissue. Adipose tissues also secrete other factors such as bone morphogenic protein (BMP), plasminogen activator inhibitor-1 (PAI-1), monocyte chemoattractant protein-1 (MCP-1), insulin-like growth factor 1 (IGF-1), insulin-like growth factor binding protein (IGFBP), fibroblast growth factor (FGF), and reactive oxygen species (ROS).

**Figure 4 biomedicines-11-02204-f004:**
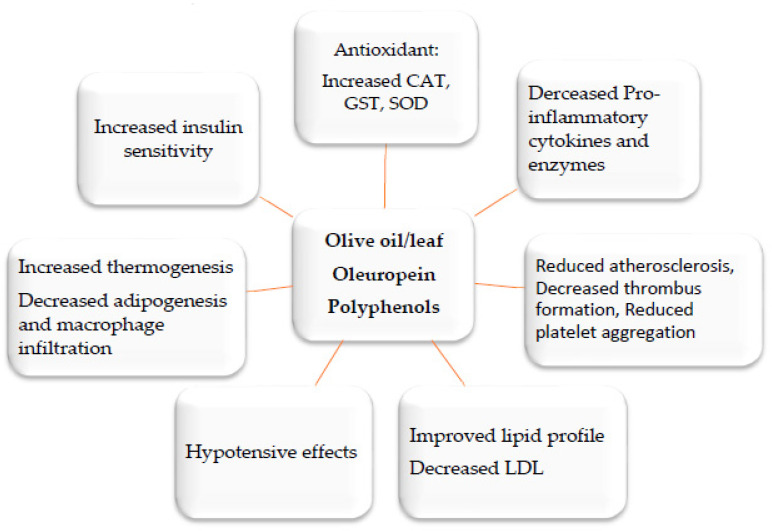
Potential synergistic anti-obesity molecular multi-target of olive oil/leaf and their active compounds oleuropein and polyphenols [28,71,72].

**Figure 5 biomedicines-11-02204-f005:**
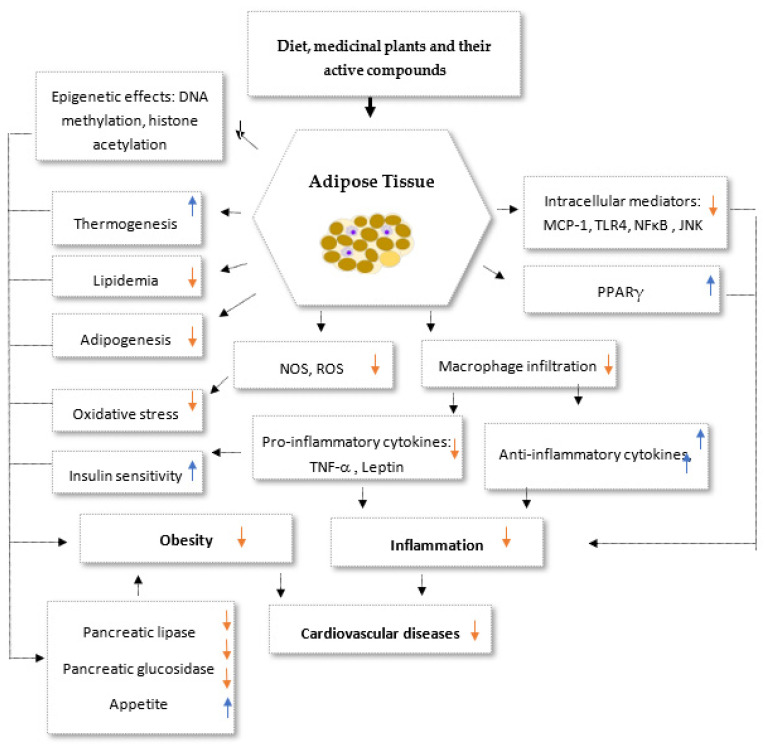
Potential synergistic anti-obesity molecular multi-target of medicinal plants and their secondary metabolites. PPARγ: Peroxisome proliferator-activated receptors gamma, NOS: nitric oxide synthase, ROS: Reactive oxygen species, TNF-α: Tumor necrosis factor alpha, TLR4: Toll-like receptor 4, MCP-1: Monocyte Chemoattractant Protei-1, NF-κB: Nuclear Factor-κB. Up arrows: incresesed effects; down arrows: decresed effects.

## Data Availability

Not applicable.

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
