# Peer review of "Management of Obesity-Related Inflammatory and Cardiovascular Diseases by Medicinal Plants: From Traditional Uses to Therapeutic Targets"

_biomedicines, 2023, doi:10.3390/biomedicines11082204_

Round 1

Reviewer 1 Report

In the manuscript entitled “Management of obesity-related inflammatory and cardiovascular diseases by medicinal plants: From traditional uses to therapeutic targets”, author has collected data on the therapeutic effects of medicinal plants/phytochemicals in treating obesity and its related inflammation and CVD diseases through an exhaustive literature and electronic database review. Author also included data from cellular and molecular mechanisms, cytokines, signal transduction cascades, and clinical trials. Inflammation is the first step in the development of cardiovascular disease and the immune response and tissue repair are coordinated by the interactions between inflammatory cytokines. Recent research has shown that certain medicinal plants and phytochemicals can treat and prevent inflammation and obesity. This review has summarized the herbal remedies for treating obesity-related inflammation and cardiovascular disease. Considering the quantity and quality of the work presented herein, I would like to ask author to incorporate below raised points in the revised manuscript. I hope, the following comments might help the author to improvise the overall manuscript.  

1, Authors should check for grammatical errors in the manuscript.

2, Authors should cross-check all abbreviations in the manuscript. Initially, define in full name followed by abbreviation.

3, Figure 1 quality is not good for publication. Author may improve the quality.

4, I would like that the conclusions fit the manuscript objectives, however the future prospective of the study should be added.

5, In order to emphasize the study's summary, at least one graphical illustration may be included.

6, Author should cite some literature to put emphasis on this interesting topic (PMID: 36093172, PMID: 35694805; PMID: 36337927; PMID: 35566252; PMID: 29901145 etc).

grammatical errors in the manuscript may be rectified 

Author Response

Dear Reviewer,

thank you for providing me with the opportunity to revise my manuscript and for the valuable comments from the reviewers. I have carefully considered and addressed all comments and suggestions in my revision. I am open to further revisions if you have any additional requests or suggestions.

Below is my point-by-point response to your comments:

1, Authors should check for grammatical errors in the manuscript.

Done

2, Authors should cross-check all abbreviations in the manuscript. Initially, define in full name followed by abbreviation.

Done

3, Figure 1 quality is not good for publication. Author may improve the quality.

 Done

4, I would like that the conclusions fit the manuscript objectives, however the future prospective of the study should be added.

I added this information in the concluding remarks

5, In order to emphasize the study's summary, at least one graphical illustration may be included.

I added this information in Figure 2. “Antiobesity and anti-inflammatory action mechanisms of medicinal plants and their impact on cardiovascular diseases”.

6, Author should cite some literature to put emphasis on this interesting topic (PMID: 36093172, PMID: 35694805; PMID: 36337927; PMID: 35566252; PMID: 29901145 etc).

PMID: 35694805; PMID: 36337927; PMID: 35566252 have been cited in the text

Reviewer 2 Report

Dear authors

This review is based on the application of plant's medicine in CDV and obesity related diseaes.

The topic is of some interest for a broad range of audience and well suited for this journal. Please be more precise with english language, look at typos and gramma allover the text.

It could be useful to add more figures since the reader could be helped by a fluent reading.I suggest to give more details on obesity mechanism as well.

References shoild be uploaded, most of all are of more than 3 years ago.

Since recent literature shows interesting data on this aspect, I suggest the following: "Food-inspired peptides from spinach Rubisco endowed with antioxidant, antinociceptive and anti-inflammatory properties", "A Study on Chemical Characterization and Biological Abilities of Alstonia boonei Extracts Obtained by Different Techniques", "An overview on plants cannabinoids endorsed with cardiovascular effects".

Look above

Author Response

Dear reviewer,

thank you for providing me with the opportunity to revise my manuscript and for the valuable comments from the reviewers. I have carefully considered and addressed all comments and suggestions in my revision. I am open to further revisions if you have any additional requests or suggestions.

Below is my point-by-point response to the reviewers’ comments:

This review is based on the application of plant's medicine in CDV and obesity related diseaes.

The topic is of some interest for a broad range of audience and well suited for this journal. Please be more precise with english language, look at typos and gramma allover the text.

Done

It could be useful to add more figures since the reader could be helped by a fluent reading. I suggest to give more details on obesity mechanism as well.

I added two new figures

References shoild be uploaded, most of all are of more than 3 years ago.

I added four new references

Since recent literature shows interesting data on this aspect, I suggest the following: "Food-inspired peptides from spinach Rubisco endowed with antioxidant, antinociceptive and anti-inflammatory properties", "A Study on Chemical Characterization and Biological Abilities of Alstonia boonei Extracts Obtained by Different Techniques", "An overview on plants cannabinoids endorsed with cardiovascular effects".

I appreciate your suggestion. However, after reviewing the paper, I’ve think that it falls outside the scope of this review article.
